# The Involvement of Neutrophil in the Immune Dysfunction Associated with BVDV Infection

**DOI:** 10.3390/pathogens12050737

**Published:** 2023-05-20

**Authors:** Karim Abdelsalam, Radhey S Kaushik, Christopher Chase

**Affiliations:** 1Department of Veterinary and Biomedical Sciences, South Dakota State University, Brookings, SD 57007, USA; 2Faculty of Veterinary Medicine, Department of Virology, Zagazig University, Zagazig 44519, Egypt; 3Department of Biology and Microbiology, South Dakota State University, Brookings, SD 57007, USA

**Keywords:** BVDV, neutrophil, macrophage, immune dysfunction, indirect effect, BVDV vaccine

## Abstract

Bovine viral diarrhea virus (BVDV) induces immune dysfunction that often results in a secondary bacterial infection in the infected animals. The underlying mechanism of BVDV-induced immune dysfunction is not well understood. The role of BVDV-infected macrophage-secreted factors was investigated. BVDV-infected monocyte-derived macrophage (MDM) supernatants down-regulated the expression of neutrophil L-selectin and CD18. Regardless of the biotype, phagocytic activity and oxidative burst were downregulated by BVDV-infected MDM supernatants. However, only supernatants from cytopathic (cp) BVDV down-regulated nitric oxide production and neutrophil extracellular traps (NET) induction. Our data suggested that BVDV-induced macrophage-secreted factors caused immune dysfunction in neutrophils. Unlike lymphocyte depletion, the negative impact on neutrophils seems to be specific to cp BVDV biotype. Interestingly the majority of modified live BVDV vaccines are based on cp strain of BVDV.

## 1. Introduction

BVDV is an endemic viral disease of cattle in North America and a major pathogen in bovine respiratory disease complex (BRDC) worldwide. BVDV infection is characterized by immune dysfunction resulting in secondary bacterial infection and consequent severe economic losses [1,2,3]. 

Several studies have demonstrated that BVDV affects the adaptive immune response [4], however, the effect of BVDV on innate immunity, especially on neutrophils, has not been well characterized. Neutrophils are the most abundant white blood cells (WBC) and play a critical role in defending the host against bacterial infections [5]. The ability of neutrophils to migrate, engulf, and kill the invading bacteria is critical in clearing the bacterial infection from the body [6]. The neutrophil immune response depends on their surface maker expression, especially CD14, CD18, and L-selectin. While CD14 helps in bacterial recognition and engulfment, CD18 and L-selectin help in chemotaxis and migration of neutrophils to the site of infection [7,8,9,10]. Previous research by our group demonstrated that BVDV infection decreased surface marker expression on innate immune cells, including neutrophils and macrophages, while decreasing phagocytic, microbicidal activity, and migration [11,12,13,14]. It was also found that BVDV-infected macrophage is the key to the immune dysfunction associated with BVDV by secreting certain mediators that manage to induce lymphocyte apoptosis in vitro. Interestingly, BVDV infection of MDM in vitro did not produce infectious progeny, and the cells did not undergo apoptosis or cytopathic effect upon infection [12,13]. Vaccination is the main strategy to control BVDV infection in the US. While most BVDV field infections are caused by non-cytopathic (ncp) biotype of BVDV, almost all commercially modified live BVDV vaccines are cp biotype to be safe to use in pregnant animals and to prevent the creation of persistently infected (PI) calves [15]. 

In the current study, we investigated the mechanism of immune dysfunction associated with BVDV by testing the effect of different biotypes of BVDV-infected macrophages’ secreted factors on neutrophil viability and functional activity. 

## 2. Materials and Methods

### 2.1. Animal 

Healthy Holstein BVDV-free heifers approximately 8 months of age were housed at the South Dakota State University dairy farm (SDSU), Brookings, SD, USA, and used in this study. The SDSU Institutional Animal Care and Use Committee (IACUC) approved animal handling and blood collection. The animals were confirmed as BVDV-free by PCR and immune-peroxidase assay on blood and ear notches, respectively [16]. 

### 2.2. Virus

Three strains of BVDV, 2 ncp and 1 cp, were used in this study (Table 1) [17]. 

### 2.3. Isolation of Monocytes from Peripheral Blood Mononuclear Cells (PBMCs) 

The isolation of PBMCs was done as previously described with modifications [12]. Briefly, whole blood was collected from the calves’ jugular vein using a sterile syringe containing 1 mL of 1000 U/mL heparin sulfate (Sigma, Sigma Chemicals, St. Louis, MO, USA). The heparinized blood was diluted 1:1 with PBS and overlaid on SepMate™ 57 lymphoprep 50 mL tubes (Stemcell Technologies, Cambridge, MA, USA) gradients and centrifuged at 1200× *g* for 20 min at room temperature (RT) using a Beckman J6-MI centrifuge. After centrifugation, the plasma and buffy coat layer were poured into a clean 50 mL conical tube (Falcon, Oxnard, CA, USA). The buffy coat was washed 5 times with PBS followed by centrifugation at 120× *g* for 10 min at RT. The viability of PBMCs was determined by trypan blue exclusion assay using 0.4% trypan blue stain [18]. The following formula was used to calculate the cell viability of the monocytes: Cell viability% = number of viable cells (unstained cells) − trypan blue-stained cells) × 100/Total counted cells. The PBMCs were resuspended in complete RPMI 1640 medium (GE Healthcare, Hyclone Laboratories, Logan, UT, USA) supplemented with 10% FBS, penicillin (100 U/mL), and streptomycin (100 μg/mL) to achieve a final concentration of 10^6^ cells/mL. The monocytes were isolated from the total PBMCs by the plastic adhesion method according to [12] with modifications. Twenty ml of suspended PBMCs were incubated in T175 Flask (Corning, NY, USA) for 3 h at 37 °C. Following incubation, the floating cells were discarded, and flasks were washed 5 times with PBS, and the wash was discarded. The adherent cells were detached with Accutase™ (eBioscience, San Diego, CA, USA). Detached cells were washed two times by suspending them in PBS and pelleting by centrifugation at 120× *g* for 10 min at RT. A sample from detached cells was submitted to Hema true blood analyzer (Heska Corp., Loveland, CO, USA), and the detached cells were identified as monocytes with a characteristic kidney shape nucleus by H&E stain. 

### 2.4. Development of Monocytes-Derived Macrophages (MDM) 

Isolated monocytes were cultured as previously described with modifications [19]. Briefly, the adherent monocytes were cultured in RPMI 1640 (GE Healthcare, Hyclone Laboratories, Logan, UT, USA) supplemented with 10% FBS, penicillin (100 U/mL), and streptomycin (100 μg/mL) at a final concentration of 1 × 10^5^ cells/well in 48-well plate followed by incubation for 7 days at 37 °C in CO_2_ incubator. The cells were fed every other day simply by replacing half of the conditioned media with fresh complete RPMI. On the 7th day, the MDM phenotype was determined [12]. Briefly, MDM were detached using Accutase™ (eBioscience, San Diego, CA, USA), washed with PBS, and pelleted by centrifugation 1000× *g* for 10 min at RT. The number of cells was adjusted to 10^6^/mL. Four primary mouse mAb antibodies for MHC-I (H58A), MHC-II (H42A), CD11b (MM10A), and CD14 (MM61A) (Monoclonal Antibody Center, WSU, Pullman, WA, USA) were used to characterize the cells. Cells were incubated with 50 µL of diluted primary antibodies 1:100 in PBS containing 1% FBS at 4 °C for 10 min, followed by washing with PBS. This was followed by staining with 1:1000 in PBS diluted FITC labeled anti-mouse secondary antibody (VMRD Inc., Pullman, WA, USA, containing 1% FBS, at 4 °C for 10 min. The stained cells were washed two times with PBS and were fixed with 200 µL of 1% paraformaldehyde. The percentage of cells expressing surface markers was calculated using BD Accuri™ C6 Plus Flow Cytometer (BD Biosciences, San Jose, CA, USA). The cultured MDM were >80% positive for MHCI, MHCII, CD11b, and CD14 markers..

### 2.5. BVDV Infection of MDM and Supernatant Collection & Irradiation

MDM were infected with one of 3 different strains of BVDV, 1373, 28508, or 296C [19]. Briefly, the MDM were infected at a multiplicity of infection (MOI) of 1 by diluting the virus stock with RPMI 1640 medium. MDM were washed with 1× PBS and infected with 10^5^ TCID_50_/mL of the virus strain at MOI of 1 in a final volume of 100 µL. The infected cells were incubated for 1 h and then washed again to remove the excess unbound virus, and 500 µL of complete RPMI 1640 medium was added to each well. Mock-infected cells treated with complete RPMI 1640 medium were the negative control, upon testing its supernatant on neutrophil phenotypic and functional activity, it seemed to have no impact. The BVDV-infected MDM were incubated for 48 h at 37 °C in a CO_2_ incubator. Supernatant collection and treatment were done as previously described [19]. Briefly, the BVDV-infected macrophage supernatants were collected 48 h p.i in 15 mL conical tubes. The supernatant was centrifuged at 1000× *g* for 10 min at RT to remove cellular debris. The BVDV-MDM supernatant was placed in 100 × 15 mm polystyrene sterile square Petri dishes (Falcon, Oxnard, CA, USA) and treated by UV irradiation at 15 cm distance for 20 min on ice to inactivate the virus [20]. The UV-irradiated supernatants were confirmed to be virus negative by BVDV immunoperoxidase assay [16]. The UV-irradiated supernatants were aliquoted and stored at −20 °C. 

### 2.6. Neutrophil Isolation

To isolate neutrophils, fifty (50) ml of peripheral blood was collected in 10 mL heparinized vacutainer tubes (BD, Franklin Lakes, NJ, USA) following IACUC guidelines. The neutrophils, along with red blood cells (RBCs), were separated by centrifuging the blood at 1000× *g* for 30 min at 25 °C. The plasma and buffy coat were removed and discarded. The cell pellet was placed into 15 mL conical tubes, and 10 mL of RBC lysing solution was added. The tube was gently inverted several times for 10 min to lyse the RBCs. After gentle inverting, the tube was centrifuged at 1000× *g* for 5 min at 25 °C. Supernatants were discarded, and cell pellets were washed 3 times using 10 mL of HBSS (the tubes were centrifuged at 1000× *g* for 5 min at 25 °C). The final cell pellet was suspended in 10 mL RPMI 1640 medium (MEM, Gibco BRL, Grand Island, NY, USA) supplemented with 10% BVDV-free fetal calf serum (FCS) (PPA, Pasching, Austria), sodium pyruvate, penicillin (100 U/mL) and streptomycin (100 μg/mL) (Sigma-Aldrich, St. Louis, MO, USA). Freshly collected neutrophils were examined for viability through trypan blue exclusion assay and CyQUANT™ LDH Cytotoxicity Assay. The purity of the isolated neutrophils was also tested via a Hema-true blood analyzer (Heska Corp., Loveland, CO, USA). Only freshly collected neutrophils with 95–98% purity and >90% viability, were used for further testing. 

### 2.7. Neutrophil Treatment with Infected MDM Supernatant and Viability Testing

To determine the effect of different BVDV-infected MDM supernatants on neutrophil viability, freshly collected neutrophils were suspended in RPMI 1640 medium supplemented with 10% FBS, penicillin (100 U/mL) and, streptomycin (100 μg/mL) to achieve final concentration 10^6^/mL. One ml of the cell suspension was added to each well of the 24-well plates, and the cells were allowed to attach. The media were carefully removed and replaced with 200 μL of each different MDM supernatant. Mock-infected MDM supernatant or lipopolysaccharide-treated (LPS; 2 μg/mL) (Sigma-Aldrich, St. Louis, MO, USA) neutrophils were used as negative or positive controls, respectively. Samples were collected at 0, 1, 3, or 6 h post-infection and examined for their viability using trypan blue exclusion and CyQUANT™ LDH Cytotoxicity Assay Kit (Life Technologies, Eugene, OR, USA). LDH is a cytosolic enzyme that will be released only from damaged cells (due to increased membrane permeability) to the outside medium that will convert the lactate in the medium into pyruvate in a coupled reaction that includes the reduction of NAD+ into NADH. The latter will be oxidized back to NAD+ in the presence of Diaphorase, which will lead to the reduction of water-soluble tetrazolium (INT) into a red Formazan product that can be measured in an ELISA reader at a wavelength of 490 nm. The assay was done using the manufacturer instructions. Absorbance was measured within 1 to 2 h at 490 nm and 680 nm. To determine LDH activity, the 680-nm absorbance value (background signal from the instrument) was subtracted from the 490-nm absorbance value for each sample in triplicate, and the average OD values were calculated. Methanol-treated neutrophil group was included as a positive cell death control. The following equation was used to calculate the viability% of neutrophils:Neutrophil viability% = 1 − [(sample OD value − spontaneous LDH OD value)/(Max LDH OD value − spontaneous LDH OD value)] × 100 

### 2.8. Immunostaining and Flow Cytometry of Neutrophil Surface Marker in Response to Infected MDM Supernatant

The expression of CD14, CD18, or L-selectin markers on the surface of bovine neutrophils was measured in response to different treatments with infected MDM supernatants, mock-treated or LPS (2 μg/mL) positive control according to [19]. Briefly, neutrophils from each treatment group were collected 6 h post-treatment and centrifuged for 5 min at 1500 rpm at room temperature (RT). Cell numbers were adjusted to 10^6^/mL in PBS containing 1% FBS. A 100 µL cell suspension was added to each well of round-bottom 96-well plates in triplicate. Fifty (50) µL of primary antibodies of anti-CD14 antibody (clone M-M9; VMRD Inc., Pullman, WA, USA), anti-CD18 antibody (clone BAQ30A; Kingfisher Biotechnology, St Paul, MN, USA), or anti-L selectin antibody (clone FMC46; Novus Biological, Littleton, CO, USA) were added to respective wells in triplicates. The primary antibodies used in staining were pre-diluted 1:100 in PBS containing 1% FBS. After adding primary antibodies, cells were incubated at 4 °C for 10 min followed by 2× washes with 200 µL PBS containing 1% FBS. After washing, cells were stained with FITC-labeled anti-mouse secondary antibody (VMRD Inc., Pullman, WA, USA) with 1:500 dilution in PBS containing 1% FBS at 4 °C for 10 min. After secondary antibody staining, cells were washed again for 2X, as described above. Finally, stained cells were fixed with 200 µL of 1% paraformaldehyde and analyzed by BD Accuri™ C6 Plus Flow Cytometer (BD Biosciences, San Jose, CA, USA). The result was recorded as a % expression. Gating strategy aimed to identify the live neutrophils and exclude cell debris, and identify the threshold for staining (Figure 1). 

### 2.9. Phagocytosis Activity of Neutrophils in Response to Infected MDM Supernatant 

The effect of BVDV-infected MDM supernatant on the phagocytic activity of neutrophils was examined through an *E. coli* phagocytosis assay kit (Cayman Chemical, Ann Arbor, MI, USA) according to the manufacturer’s instructions. Briefly, freshly isolated neutrophils were suspended at a concentration of 10^6^ cells/mL in a culture medium. Different infected MDM-supernatant treatments were applied to the corresponding wells in triplicates, including the 3 BVDV strain-specific MDM supernatants, the mock-treated MDM supernatant, and LPS (2 μg/mL) stimulated neutrophils as negative and positive controls respectively. Hundred (100) μL of cells of each sample in triplicate were transferred to a 96-well plate where the cells got mixed with FITC-labeled inactivated *E. coli* at a ratio of 1:10. Neutrophils were incubated at 37 °C for 1 h. To assess the degree of phagocytosis, cells were centrifuged at 400× *g* for 5 min, and the supernatant was discarded. After that, the cells were resuspended in 100 μL of trypan blue to quench the non-specific surface FITC signal. The cells were washed twice with the assay buffer, supplied with the kit, and resuspended in 200 μL of assay buffer. The flow cytometry scan was done using BD Accuri™ C6 Plus Flow Cytometer (BD Biosciences, San Jose, CA, USA). The flow cytometer records the % of fluorescent cells in each treatment. The % of fluorescent cells indicates successful phagocytosis. For analysis, the % of fluorescent cells in treated groups was contrasted with that of the mock-treated control and reported as % phagocytosis where the mock-treated was considered as 100% according to the following equation:% phagocytosis = % of fluorescent cells in the treated group/% of fluorescent cells in the mock-treated × 100. 

### 2.10. Neutrophil Oxidative Burst Activity in Response to Infected MDM Supernatant

The effect of BVDV-infected MDM supernatant on neutrophil oxidative burst activity was examined [11] with modification. Briefly, both treated and mocked treated neutrophils (for 6 h) were washed with PBS and centrifuged at 500× *g* for 10 min at RT. After washing, neutrophils were resuspended in RPMI 1640 medium supplemented with 10% FBS, penicillin (100 U/mL), and streptomycin (100 μg/mL) to achieve a final concentration of 10^6^ cells/mL. Neutrophil suspension (200 µL) from each treatment was then transferred to individual wells in a 96-well plate. Twenty (20) µL of 10 µm DHR 123 (Sigma-Aldrich, St. Louis, MO, USA) was added to each well, except DHR123 negative control cells. DHR 123 is a colorless dye which passively enters the cell and produces fluorescence in the presence of reactive oxygen species. After adding DHR 123, plates were incubated at 37 °C for 15 min. To induce oxidative burst, 50 µL (10 nM) Phorbol 12-myristate 13-acetate (PMA) was added to each well (except the designated wells for the assay negative control as well as DHR 123-negative control), and the plate was incubated for 15 min at 37 °C. After 15 min of incubation, plates were washed with PBS and fixed with 1% paraformaldehyde in PBS. The oxidative burst activity in fixed cells was measured using the BD Accuri™ C6 Plus Flow Cytometer (BD Biosciences, San Jose, CA, USA), and the result was recorded in %. The DHR 123-negative control was used to set the fluorescence gates on the flow cytometer. The oxidative burst was reported in percentage by the machine software. 

### 2.11. Nitric Oxide Activity of Neutrophils in Response to Infected MDM Supernatant 

The effect of BVDV-infected MDM supernatant on nitic oxide activity of neutrophils was examined using the Griess Reagent. Briefly, freshly collected neutrophils were treated with 3 BVDV strain-specific MDM supernatants as well as mock-treated control and incubated for 6 h at 37 °C. Cell samples from each treatment were then stimulated with LPS (25 µg/mL) (Sigma, St. Louis, MO, USA) for 1 h. Then, supernatants were collected, and nitric oxide (NO) concentration was determined using Griess reagent (Sigma, St. Louis, MO, USA) using Nitrite (NO2) measurement as an indicator of NO production according to [21]. Briefly, in each well of a 96-well plate, a 100 µL of culture supernatant was added to 100 µL of Griess reagent, which was made of a 1:1 mixture of 1% sulfanilamide and 0.5% N-(1-naphthyl) ethylenediamine dihydrochloride (Sigma, St. Louis, MO, USA) in 30% acetic acid. Reactions were performed in triplicate at RT for 10 min. Absorbance was then measured at 540 nm in a microplate reader (Biotek, ELx808, Winooski, VT, USA). Nitrite concentration was evaluated by plotting the average OD values of each sample (in triplicates) against sodium nitrite standard curve (Sigma, St. Louis, MO, USA) with a detection range of 1.95–200 μmol/mL. 

### 2.12. Neutrophil Extracellular Traps (NETs) Formation in Response to Infected MDM Supernatant 

NETS are networks of extracellular fibril matrix, primarily composed of DNA from neutrophils. Upon in vitro stimulation with PMA, neutrophils release granular proteins and chromatin to form an extracellular fibril matrix to trap bigger pathogens that are hard to be engulfed. These NETs can be measured through the quantification of extracellular DNA released in the active process. Quant-iT™ PicoGreen™ dsDNA Assay Kit (Thermo Fisher Scientific, Waltham, MA, USA) was used to measure the extracellular DNA released in the supernatant. The extracellular DNA was separated from the network with the aid of micrococcal nucleases (New England Biolabs, Ipswich, MA, USA). The DNA was stained with an asymmetrical cyanine dye, a free dye that does not fluoresce but upon binding to dsDNA exhibits a >1000-fold fluorescence enhancement. To measure the effect of infected MDM supernatant on neutrophil extracellular traps (NETs) formation, a neutrophil sample from each treatment group was pelleted 6 h post-treatment and adjusted to 5 × 10^5^/mL and washed twice with PBS before being submitted for PicoGreen assay according to manufacturer’s instructions. Briefly, the neutrophil samples (in triplicates) were incubated with 20 nM PMA to stimulate NET formation for 4–5 h. Then, the neutrophils were pelleted at 250× *g* for 3 min, and the supernatant was discarded. Micrococcal nuclease (New England Biolabs, Ipswich, MA, USA) 0.1 U/μL, in micrococcal nuclease buffer, was added to each sample pellet and incubated for 15 min at 37 °C. One μL of each sample was added to 99 μL of 1× TE buffer, supplied in the kit, in the microplate well, and mixed by pipetting up and down. A lambda DNA standard was prepared with a range of 5–2000 ng/mL in duplicates for measuring the NET formation level in each sample by plotting the sample values against the standard curve values. An equal volume (100 μL) of diluted PicoGreen 1/400 in TE buffer was added to the volume of each standard and sample and mixed by pipetting up and down. The microplate was covered with foil and incubated at RT for 2–5 min. The plate was read at excitation of 485 nm/20 nm and emission of 530 nm/25 nm wavelength/bandwidth. Another mock-treated-non-PMA-stimulated was included to convert the level of NET formation into % through the following equation:(Amount of extracellular DNA in PMA-stimulated treatment − Amount of extracellular DNA in non-stimulated mock-treatment)/Amount of extracellular DNA in PMA stimulated mock treatment − Amount of extracellular DNA in non-stimulated mock-treatment) × 100.

### 2.13. Statistical Analysis 

Data were analyzed using a paired t-test at a 5% level of significance (Microsoft EXCEL, MAC 2011) to assess the significance of the differences between mean values of treated and control samples at each time point. Every experiment was achieved using at least 3 animals (biological replicates) conducted 3 times each (technical replicates) to confirm the reproducibility of the methods used. The variations in results were calculated by standard deviation at each time point. 

## 3. Results

### 3.1. Effect of BVDV-Infected MDM Supernatant on Neutrophil Viability 

The neutrophil viability was tested up to 6 h post-treatment as the viability of mock-treated cell control decreased to 80%. The neutrophil viability of any of the BVDV-infected supernatants was not significantly different compared to the mock-treated control. The cytopathic 296C-treated neutrophils had an average viability of 88% while the non-cytopathic 1373 or 28508-5 supernatant-treated neutrophils had a viability of 87.8% and 88.2%, respectively, over 6 h compared to 88.8% viability of Mock-treated control neutrophils. Methanol-positive control showed 100% cell death by 3 h post-treatment (Figure 2). All downstream testing of neutrophils was set up to 6 h post-treatment, based on the viability finding reported from this assay. 

### 3.2. Effect of BVDV-Infected MDM Supernatant on Neutrophil’s Surface Marker Expression 

All BVDV supernatants down-regulated CD18 expression at 6 h post-treatment by 11.8% for 296C, 10.2% for 1373, and 7.6% for 28508 supernatants (*p* = 0.096–0.088). In contrast, LPS-treated control showed a significant up-regulation (*p* < 0.05) with a 13.8% increase in CD18 expression, compared to the mock-treated control (Figure 3). All BVDV supernatants significantly (*p* < 0.05) down-regulated the expression of L-selection by 14.1%, 20.6%, and 18.2% for 296c, 1373, or 28508 supernatants, respectively, at 6 h post-treatment. LPS-treated controls also down-regulated L-selection expression on the surface of treated neutrophils by 23.9% compared to the mock-treated control in a significant way (*p* < 0.05) (Figure 3). CD14 expression was low on the surface of neutrophils in all 3 viral supernatant groups and was not affected 6 h post-treatment compared to the mock-treated control, while the LPS-treated positive control group showed a significant up-regulation of CD14 expression (data not shown).

### 3.3. Effect of BVDV-Infected MDM Supernatant on Neutrophil Phagocytic Activity 

All BVDV supernatants reduced the phagocytosis ability of neutrophils at 6 h post-treatment compared to the mock-treated control (*p* = 0.14–0.17). Treatment of neutrophils with 296c, 1373, or 28508 supernatants resulted in 95%, 93.3%, and 95.5% phagocytosis, respectively, compared to 100% for mock-treated control (Figure 4).

### 3.4. Effect of BVDV-Infected MDM Supernatant on Neutrophil Oxidative Burst 

Neutrophils treated with 296c, 1373, or 28508 supernatants oxidative burst activity upon PMA stimulation of 41.6%, 43.3%, and 47.9%, respectively, compared to 50.3% in the mock-treated positive control. The difference between each treatment in contrast to the mock-treated control was insignificant (*p* = 0.058–0.076) (Figure 5).

### 3.5. Effect of BVDV-Infected MDM Supernatant on Neutrophil Nitric Oxide (NO) Activity 

Neutrophils treated with 296c, 1373, or 28508 supernatants produced 78.8, 117.3, and 119.8 µmol/mL of NO, respectively, upon stimulation with LPS compared to 127.6 µmol/mL of NO in mock-treated control. Only the cp 296c supernatant significantly (*p* < 0.05) down-regulated nitric oxide production in LPS-stimulated neutrophil compared to mock-treated LPS-stimulated control, while the ncp supernatant trended lower from the mock-treated group (*p* = 0.072–0.87) (Figure 6).

### 3.6. Effect of BVDV-Infected MDM Supernatant on Neutrophil Extracellular Trap (NET) Formation 

Neutrophils treated with 296c, 1373, or 28508 supernatants showed 81.4%, 97.6%, and 98.2%, respectively, of NET formation compared to 100% in mock-treated control. Only the cp 296c supernatant significantly (*p* < 0.05) down-regulated NET formation in PMA-stimulated neutrophils, while the ncp supernatants showed a non-significant difference from the mock-treated PMA-stimulated group (*p* = 0.096–0.124) (Figure 7).

## 4. Discussion

BVDV is an endemic viral disease of cattle in North America despite the wide variety of BVDV vaccines available on the market. The main concern associated with BVDV infection is the immune dysfunction manifested by severe lymphoid depletion and secondary bacterial infection in the field. Previously, we described the key role of macrophages in BVDV immune dysfunction. We found that the supernatant of BVDV-infected macrophages induced lymphocyte apoptosis associated with severe lymphoid depletion in the field [13]. In this study, we examined the effect of BVDV-infected MDM supernatants on neutrophil viability and functional activity. Both cp and ncp biotypes of BVDV were included in our in vitro study since most infections in the field are caused by ncp BVDV carried by a persistently or acutely BVDV-infected animal, while most of the commercially available vaccines use cp BVDV strains [15,22]. 

In the current work, supernatants from BVDV-infected MDM did not affect neutrophil viability up to 6 h post-treatment. Our findings are consistent with the in vitro findings of another study where neither cp TGAC nor ncp TGAN strains of BVDV-affected neutrophil viability at the same timepoint [11]. An in vivo study reported both neutropenia and impaired neutrophil activity following BVDV infection [23]. The difference between the in vivo study and our results could be due to a direct effect on hematopoietic production, or it could also be due to cell margination or other hemodynamic mechanisms that may falsely affect the neutrophil numbers in the blood. 

Our study demonstrated that the BVDV-infected MDM supernatants downregulated the neutrophil surface expression of both CD18, and L-selectin compared to the mock-treated control at 6 h post-treatment. This reduction in expression was only statistically significant for L-selectin. While CD18 regulates neutrophil maturation and release to circulation, L-selectin facilitates margination and signal trafficking to the site of local infection [9]. Impairment of their expression may affect the overall functional activity of neutrophils [24,25]. Another study reported that both L-selectin and CD18 were downregulated in response to direct infection of neutrophils with BVDV strains that consequently impact neutrophil migration and functional activity [11,26]. CD14 expression was also evaluated in the current study. However, its initial expression was low on the mock-treated neutrophil (<13% at 0 h), and no significant difference was observed between groups at 6 h post-treatment. CD14 recognizes endotoxin on invading gram-negative bacteria as well as its cooperative role with TLR-4 in phagocytosis. We previously found that high virulent BVDV-infected-MDM supernatant significantly downregulated the expression of CD14 on the surface of macrophages as early as 12 h post-treatment. CD14 downregulation was also associated with the reduction in the phagocytic activity of macrophages in the same study [13] and of neutrophils in hepatitis C virus (HCV) patients [27]. 

The immune response to invading pathogens is divided into innate and adaptive immune responses. Innate immune response is general and quick and includes many barriers and cells. Neutrophils are the most abundant and key cells of the innate immune response to protect the body against bacterial infection. Impairment of neutrophil activity will consequently compromise the body’s defense against opportunistic bacteria. One of the main consequences of BVDV infection in the field is secondary bacterial infection which increases morbidity and mortality. In the current study, we evaluated different functional activities of neutrophils in response to different BVDV-infected MDM supernatants. Phagocytosis is one of the most effective defense mechanisms that is used by neutrophils and macrophages to defend against microbial invasion. Phagocytosis is the link between innate and adaptive immune responses as phagocytic activity leads to antigen degradation and presentation to the naïve lymphocytes and initiation of the adaptive immune response. In this study, the BVDV-infected MDM supernatants, no matter the biotype or virulence, downregulated the phagocytic activity of neutrophils at 6 h post-treatment compared to the mock-treated control. Neutrophils isolated from cp BVDV-infected animals had reduced oil droplet uptake compared to neutrophils isolated from non-infected control [23]. Another study reported that only virulent BVDV or its corresponding infected MDM supernatants significantly inhibit the phagocytic activity of treated MDM compared to the mock-treated control as early as 12 h post-infection [13]. One reason for our phagocytic reduction not being significant is that the experiment was conducted for up to 6 h post-treatment. This time period could be too early for the MDM supernatants to show their full impact on phagocytic activity. Previous research has reported that cp BVDV strains were able to reduce neutrophil phagocytic activity [11,26]. The difference could be either due to the use of different strains [26] or that our findings are based on the effect of infected MDM supernatant rather than direct infection with the virus. 

Oxidative burst, an important of microbial killing by neutrophils, was also affected. Both cp and ncp BVDV-MDM supernatants trended to reduce the oxidative burst activity of PMA-stimulated neutrophils compared to the mock-treated PMA-stimulated control. The cp MDM supernatant induced the most downregulation with a 7.2% decrease at 6 h post-treatment. Direct infection of neutrophils with different BVDV strains, regardless of their virulence or biotype, downregulated oxidative burst activity at 6 h post-infection [11]. The difference between this finding and ours could be because we tested the infected MDM supernatants rather than directly infecting neutrophils with BVDV strains. Inhibition of neutrophil oxidative burst following BVDV infection has also been reported in other studies that have been conducted on neutrophils isolated from vaccinated animals with modified live cp BVDV-based vaccine [28]. 

Nitric oxide production, another effective neutrophil defense mechanism, was also affected. Only cp BVDV supernatant significantly reduced nitric oxide production in the stimulated neutrophils. Cp 296c supernatant reduced nitric oxide production by 37%, while ncp strains had non-significant statistical reduction. Oxidative burst and NO production are the two major mechanisms by which the phagocytosed pathogen will be destroyed. Our data was different from previous work where ncp MDM supernatants did not significantly affect NO production at 12–24 h post-treatment [13]. Other studies showed that infection with HCV, another flavivirus, upregulated NO production in mice livers [29]. 

NET formation, another important innate defense mechanism, was also affected. Cp MDM supernatant significantly downregulated NET formation by 18.6%, while the ncp MDM supernatants had less than 3% inhibition. The infection of neutrophils with BVDV led to significant up-regulation in NET formation, regardless of their virulence or biotypes [11]. The difference between this finding and ours could be due to the different design of the experiment, where we examined the effect of the BVDV-infected MDM supernatants rather than BVDV direct infection of neutrophils. Another difference could be the way that the NET formation was based on the detection of the extracellular DNA rather than detecting extracellular elastases [11]. Another important difference was PMA activation as we examined the impact of the infected supernatants on the NET formation from the PMA-stimulated neutrophils, while the other study examined the ability of BVDV infection of neutrophils to induce NET formation in non-stimulated cells where PMA was used only as a positive control. 

Further research is still needed to identify the content of the infected macrophage’s supernatant that mediated the associated immune dysfunction. Previous study by our group excluded the involvement of secretory viral protein and certain pro-inflammatory cytokines [13], however, potential other factors need to be investigated in the next step of this research. 

## 5. Conclusions

Despite vaccination, BVDV is still endemic in North America. BVDV is one of the major pathogens in the BRDC and is associated with lymphoid depletion and secondary bacterial infection in infected animals. Neutrophils are the main player in defending the body against invading and opportunistic bacteria. In this study, the effect of BVDV-infected MDM supernatants on neutrophil viability and functional activity was evaluated. Our data supported the key role of the macrophage in the immunopathogenesis of BVDV infection. Our results provided a possible explanation for the immune dysfunction associated with BVDV infection that is manifested by secondary bacterial infection in the field. Our findings also shed light on the possible impact of cp BVDV, which is widely used in commercial BVDV vaccines in North America, in downregulating neutrophil functional activity. The impairment of neutrophil activity caused by BVDV, especially cp strain, could be a possible explanation for the secondary bacterial infection associated with BVDV in the field. Future studies comparing the impact of field versus vaccinal strain of BVDV on macrophage and neutrophil activity are needed to exclude the involvement of the current vaccine strategy in the endemic nature of BVDV in North America. 

## Figures and Tables

**Figure 1 pathogens-12-00737-f001:**
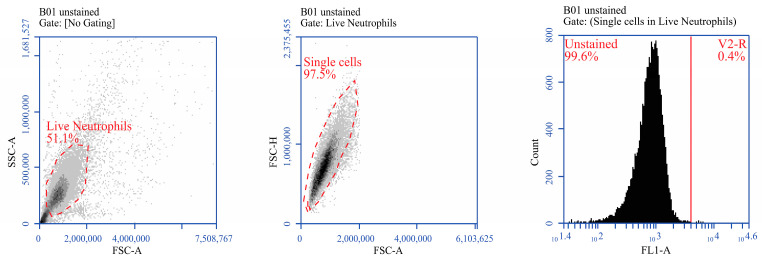
Gating strategy for bovine neutrophils: The cells were first gated on forward scatter (FSC-A) and side scatter (SSC-A) plot to identify the live neutrophil and exclude the cell debris. Then single cells were gated on FSC area (A) and height (H). This was followed by setting the gate for staining by setting the threshold (red vertical line) on unstained cells.

**Figure 2 pathogens-12-00737-f002:**
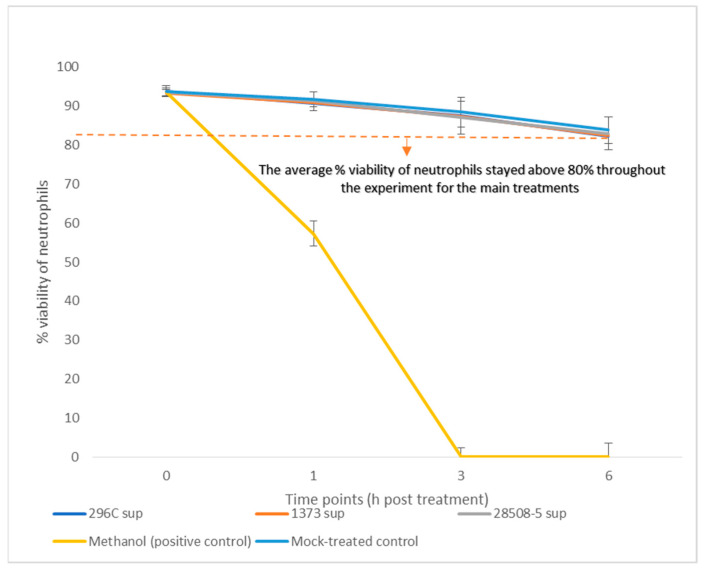
The effect of BVDV-infected MDM supernatants on neutrophil viability: % Viability of neutrophils in response to treatment with BVDV-infected MDM supernatant (296C, 28508, and 1373 sup) or methanol-treated death control in contrast to mock-treated (cells + medium only) control. The result is the average of at least 3 replicates *n* = 3.

**Figure 3 pathogens-12-00737-f003:**
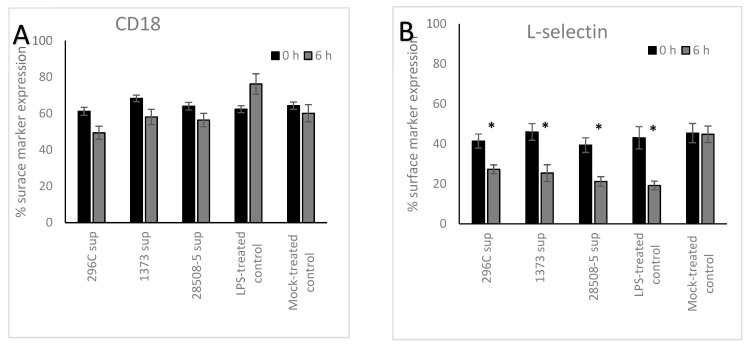
The effect of BVDV-infected MDM supernatants (296C, 1373, and 28508-5 sup) on neutrophil surface marker expression: (**A**) CD18, (**B**) and L-selectin. Sup: supernatant *: *p* < 0.05 (>95% confidence). The result is the average of at least 3 replicates (*n* = 3).

**Figure 4 pathogens-12-00737-f004:**
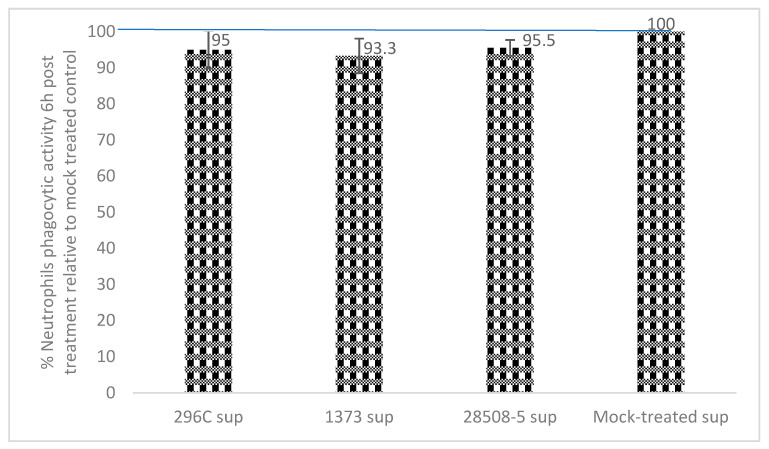
The average phagocytic activity of neutrophils in response to treatment with different BVDV-infected MDM supernatants (296C, 1373, and 28508-5 sup) relative to the mock-treated (non-treated normal neutrophils engulfing bacteria) control. *n* = 3.

**Figure 5 pathogens-12-00737-f005:**
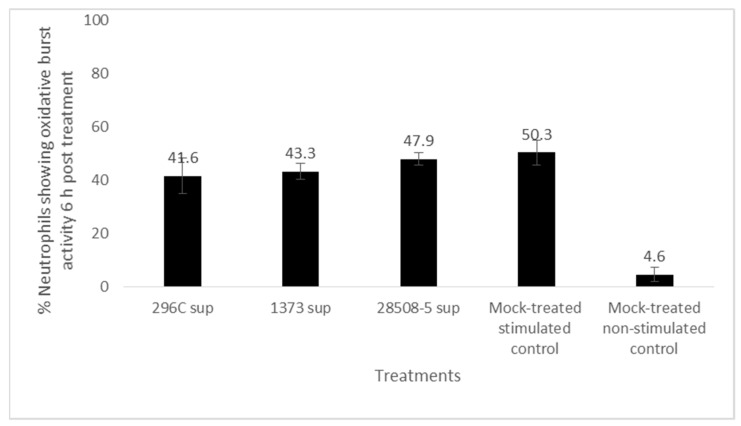
The effect of BVDV-infected MDM supernatants (296C, 1373, and 28508-5 sup) on PMA-stimulated neutrophils’ oxidative burst activity compared to the mock-treated stimulated (non-treated, PMA stimulated) control. Sup: Supernatant (*p* > 0.05). The result is the average of at least 3 replicates (*n* = 3).

**Figure 6 pathogens-12-00737-f006:**
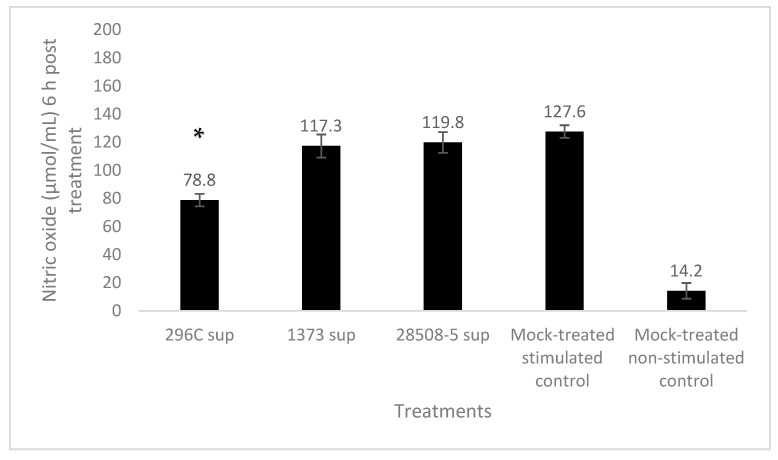
The effect of BVDV-infected MDM supernatants on neutrophil nitric oxide production 6 h post-treatment compared to the mock-treated control. *: significant (*p* < 0.05) Sup: Supernatant. The result is the average of at least 3 replicates (*n* = 3).

**Figure 7 pathogens-12-00737-f007:**
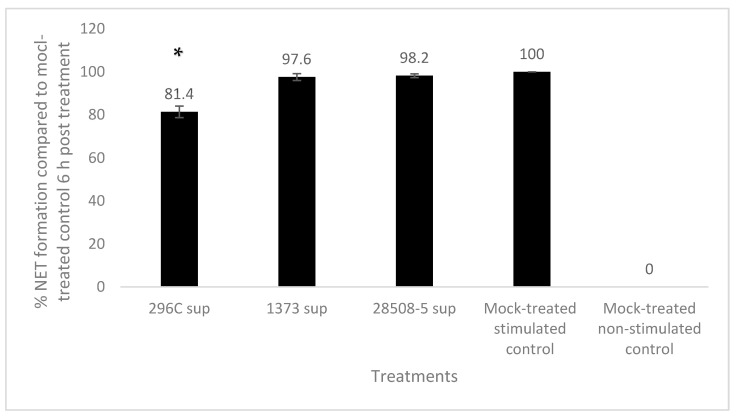
The effect of BVDV-infected MDM supernatants (296c, 1373, and 28508-5 sup) on induction of NET formation in PMA-stimulated neutrophils 6 h post-treatment compared to the mock-treated, PMA-stimulated control. *: significant (*p* < 0.05) Sup: Supernatant. The result is the average of at least 3 replicates (*n* = 3).

**Table 1 pathogens-12-00737-t001:** Different strains of BVDV used in the current study.

BVDV Strains	Biotype	Virulence	Titer
1373	ncp, BVDV-2	High virulence	5.8 × 10^5^
28508-5	ncp, BVDV-2	Low virulence	7.2 × 10^6^
296C	cp, BVDV-2	Virulent	5.2 × 10^5^

## Data Availability

Data is included within the article. The datasets used during the current study are available from the corresponding author upon request.

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
