# Peer review of "The Involvement of Neutrophil in the Immune Dysfunction Associated with BVDV Infection"

_pathogens, 2023, doi:10.3390/pathogens12050737_

Round 1

Author Response

Dear Reviewer 1, Thank you for your review. Please see the attachment.

Reviewer 2 Report

This manuscript presents an interesting in-vitro study on the effects of BVD virus on the innate immune system.

Throughout the manuscript there are jumps between past and present tense - that should be corrected (either way) throughout for readability (line 388 vs 390 for example). The manuscript is also a bit North American-centric - e.g. line 27 - BVD actually affects cattle populations around the world. There are also some parts of the world that use inactivated vaccines, so line 46 could do with a clarification.

My biggest concern with the experimental design is the mention in line 57-58 of the testing for BVD freedom in the blood donors - surely you didn't just test for AB, arguably with not the most sensitive technique either, but also for virus or viral antigen (or if not, that needs to be explained).

The effects observed in the assays are largely not that great, when compared to the controls and there should be some caveat about the clinical significance. Also, there should be some discussion regarding in vitro results versus in vivo significance.

Author Response

Dear Reviewer 2, Thank you for your review and comments. Please see the attachment. 

Round 2

Reviewer 2 Report

Thank you for addressing my concerns - all good from my side now

Author Response

Thank you very much for your comments.